# Involving the headteacher in the development of school-based health interventions: A mixed-methods outcome and process evaluation using the RE-AIM framework

**Danielle L. Christian** [1]☯*, **Charlotte Todd**[2]☯, **Jaynie Rance**[3]‡, **Gareth Stratton**[4]‡, **Kelly A. Mackintosh**[4]‡, **Frances Rapport**[5]‡, **Sinead Brophy**[2]☯

1 Faculty of Health and Wellbeing, University of Central Lancashire, Preston, United Kingdom, 2 Swansea University Medical School, Swansea University, Swansea, United Kingdom, 3 College of Health and Human Science, Swansea University, Swansea, United Kingdom, 4 College of Engineering, Swansea University, Swansea, United Kingdom, 5 Australian Institute of Health Innovation, Macquarie University, Sydney, Australia

☯ These authors contributed equally to this work.
‡ These authors also contributed equally to this work.
* dchristian@uclan.ac.uk

**Data Availability Statement:** The data from this research study are not publicly available due to participant confidentiality. Data from this research

## Abstract

Although interventions delivered in school settings have the potential to improve children's health and well-being, the implementation of effective interventions in schools presents challenges. Previous research suggests facilitating greater autonomy for schools to select interventions aligned to their needs could improve implementation and maintenance. The aim of this mixed-methods outcome and process evaluation was to explore whether involving headteachers in the developmental stages of health interventions influenced adoption, effectiveness (e.g. pupil fitness and physical activity, assessed quantitatively), implementation and maintenance (assessed quantitatively and qualitatively).

Three UK primary schools were provided with a choice of five evidence-based physical activity interventions: Playground scrapstore, daily classroom refreshers, alternative afterschool clubs, parent and child afterschool activities and an 'In the Zone' playground intervention. To evaluate the impact of this autonomous approach, semi-structured interviews with headteachers (n = 3), teachers (n = 3), and a private coach, and focus groups with pupils aged 9–11 (n = 6, 31 pupils, 15 boys), were undertaken. This was alongside an outcome and process evaluation, guided by the RE-AIM framework. This study assessed the impacts on adoption, implementation and maintenance of the autonomous approach and the effect on physical activity (seven day accelerometry–GENEActiv) and aerobic fitness (20m shuttle run). All three schools adopted different intervention components; alternative afterschool clubs, parent and child afterschool activities and daily classroom refreshers. Headteachers welcomed greater autonomy in developing school-based interventions and appreciated the more collaborative approach. Mixed results were reported for the effectiveness, implementation and maintenance of the interventions adopted. Allowing pupils choice and promoting a positive school environment were key factors for enhancing engagement.

study contain information that are identifiable at both the school and individual level. Ethical approval was granted by the Swansea University College of Human and Health Sciences Research Ethics Committee, provided participants' data was only accessible by the research team. Participants did not consent to having their data publicly available. Requests for access to the data may be directed to the Swansea University College of Human and Health Sciences Research Ethics Committee by emailing CHHS-Ethics@swansea.ac. uk.

**Funding:** The work was funded by the Public Health Wales Swansea Healthy City Programme (SB received award). Additionally, funding was in conjunction with support from The National Centre for Population Health and Wellbeing Research (https://ncphwr.org.uk/). The funders had no role in study design, data collection and analysis, decision to publish, or preparation of the manuscript.

**Competing interests:** The authors have declared that no competing interests exist.

Moreover, promoting inclusive physical activity projects with a consideration of existing curriculum pressures aided implementation. This mixed-methods study provides valuable insights about autonomous approaches to inform further development, implementation and maintenance for future interventions.

## Introduction

Physical activity has been positively associated with both physiological and psychosocial health [1]. Current guidelines recommend that children engage in at least 60 minutes moderate-to-vigorous physical activity (MVPA) every day [2], yet few children engage in sufficient levels to meet these guidelines [3, 4]. Given that physical activity behaviours have been shown to track into adulthood [5], physical activity-promoting interventions implemented during childhood are imperative. Additionally, physical activity is known to decrease from childhood to adolescence [6, 7], with the transition from primary to secondary school marking a critical period for intervention.

Schools have been identified as an appropriate setting for such approaches [8] and many physical activity interventions have been shown to be effective in primary school settings [9–11]. However, it has been argued that only modest effects have been observed [12]. Whilst non-curricular approaches, such as playground interventions, afterschool sessions and daily classroom refreshers hold some promise under intervention conditions, the translation of effective research findings to the school in a 'real world' setting can be problematic [13]. Previous formative research has identified that providing headteachers with greater autonomy to select suitable interventions to align with their specific school's needs and facilitate contextual adaptations could improve implementation and maintenance [14–16]. Guidelines for designing complex interventions suggest that permitting schools an element of local adaptation enables interventions to more closely align with their target population [17]. Moreover, the 'Health Promoting Schools' agenda recommends allowing schools more choice in creating their own holistic, health-centred environment that endorses their individual values and ethos [18]. Despite these guidelines and recommendations, there are few established health interventions which allow headteachers a choice of autonomy over different types of intervention. Specifically, the Action Schools! BC (AS!BC) choice-based project, implemented across Canada, has demonstrated popularity with teachers, pupils and Governmental parties alike [19], despite demonstrating little long-term effectiveness; especially for boys [20]. The AS!BC intervention is composed of six 'Action zones' including school environment, scheduled P.E., classroom action (mandatory), family and community, extra-curricular and school spirit. Despite designs such as the AS!BC, there remains a paucity of research where headteachers have complete autonomy over their school's interventions, and the popularity of the choice-based approach of the AS!BC framework warrants further exploration. Therefore, the aim of the present study was to involve headteachers in the developmental stages of school-based health interventions to allow them greater autonomy and explore how this influenced adoption, effectiveness, implementation and maintenance.

## Methods

### Recruitment

Nine primary schools in South Wales were contacted to participate in the Community Led Active Schools Programme (CLASP). Deprivation was classified to assess the socioeconomic

variability using individual free school meal entitlement [21], with free school meal eligibility (FSM) ranging from 9% to 53% (mean 37.5%). These nine schools were selected as they had participated in the formative phases of the intervention [14, 15], and three expressed an interest in continued participation. These three headteachers were provided with a project description and following an expression of interest, a further meeting was set-up to discuss participation. All children in Year 5 and 6 (aged 9–11 years) at participating schools were eligible for participation within the study. Of the 125 children eligible, informed parental consent and participant assent forms were returned by 85 children (44 boys, 41 girls, 68% response rate).

## Ethical approval

Ethical approval was granted by the Swansea University Research Ethics Committee. Written informed consent was obtained from headteachers, teachers and the external coach prior to participation in the interviews. Written informed parental consent and child assent was obtained prior to participation in the research components (e.g. focus groups). Parental consent forms were also required for participation in afterschool sessions.

## Intervention components

All three headteachers were presented with a choice of five evidence-based physical activity intervention components (Table 1), focusing on different school periods. Headteachers were asked to consult with key members of staff to discuss which components would best suit their school needs. The final selection regarding which components to implement (one or two) occurred during a face-to-face consultation/interview with the research team. All five were free to the schools and pupils, all costs were covered through CLASP, and teachers were provided with an overview of the how their chosen interventions should be implemented.

Table 1. Intervention components with descriptions and supporting evidence.

| Intervention components | Description | Supporting evidence |
|---|---|---|
| Daily classroom refreshers | 10-minute bouts of physical activity to break up sedentary time. Physical activity card ideas issued to school staff, with teachers encouraged to allow children to take greater ownership regarding the design and delivery of their own activities. | [22–26] |
| Alternative activities | Alternative activities, such as street dance and skateboarding (chosen by pupils themselves), were promoted afterschool and led by an external, private coach. | [27–30] |
| Parent and child afterschool sessions | Combined parent and child afterschool sessions can improve enjoyment and reduce the need for child care; a barrier to physical activity for parents. This included activities such as family boxfit and was led by a private coach. | [31–35] |
| Playground Scrapstore | The Playground Scrapstore provided clean, safety-checked scrap equipment (e.g., cardboard boxes, tubes, cable reels) to promote imaginative free-play during playground breaks. Additional loose games equipment during break times has been shown to improve physical activity. | [36–44] |
| 'In The Zone' project | 'In the Zone' project encouraged the playground to be divided more fairly to encourage active play whilst enabling more organised, structured playtimes. An interactive DVD resource pack was provided as well as a training workshop for lunchtime supervisors. | [37, 38, 45–48] |

## Intervention design

Baseline quantitative measurements were taken over a two-week period (January), in addition to 1:1 interviews with headteachers (mean 18 minutes, range 15–21 minutes) to select intervention choices. All three schools then underwent their individual interventions for three months, followed by a two-week post-intervention measurement period (April). Follow-up measurements were performed three months after post-intervention (July) to assess maintenance of the project and any consequent change in health behaviours, again over a two-week period (Fig 1). For reference, the UK school structure runs from September to July. All measurements were undertaken during school time.

**Qualitative measures.** Semi-structured interviews were conducted with headteachers post-intervention (mean 22 minutes, range 14–24 minutes) and again at follow-up (mean 29 minutes, range 21–34 minutes) to ascertain views on the provision of greater autonomy with respect to school-based health interventions (Fig 1). Interviews provided the opportunity to obtain a richer, more in depth understanding regarding participants' views of the implementation fidelity and maintenance [49]. All interviews were conducted individually in headteachers' offices and an open-ended question-based topic guide was used throughout to facilitate discussion. Two experienced researchers (DC & CT) were present at each interview; one facilitated the interview, while the other noted key points, as well as researcher and participant interactions. The second researcher also reported back a brief summary of the interview to participants at the end of the interview, to ensure respondent validation [50]. All interviews were digitally recorded and transcribed verbatim. Following each interview, both researchers debriefed and adapted the topic guide accordingly for the next, incorporating tenets of an iterative, inductive approach to build a framework for thematic analysis; a methodology detailed elsewhere [51, 52]. At post-intervention, semi-structured interviews were also conducted individually with the Year 5/6 teachers, or deputy headteachers, at all schools (mean 13 minutes,

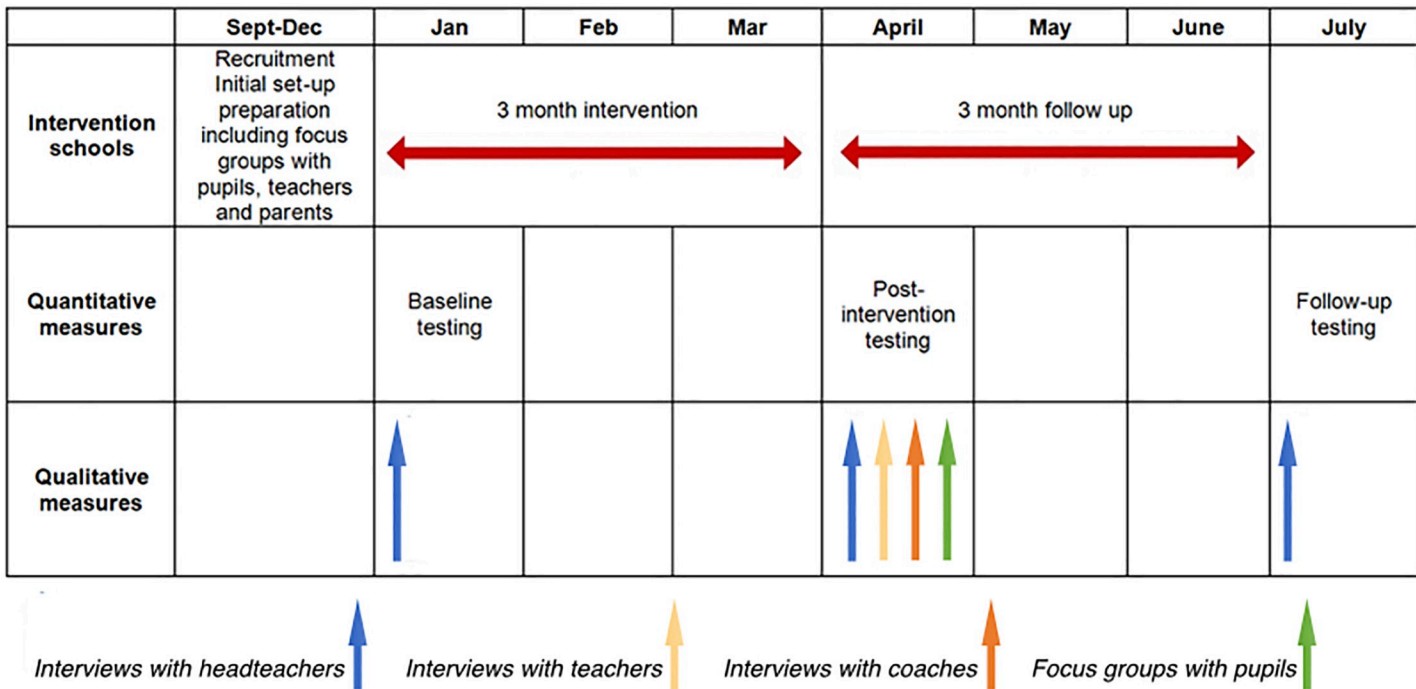

**Fig 1. CLASP intervention timeline.**

range 11–16 minutes), and one private coach who had undertaken sessions as part of the intervention (25 minutes). This was to explore intervention implementation in greater detail. The two other coaches declined an invitation to participate in an interview due to work commitments. No additional funding was provided for their participation in interviews, so as not to incentivise their involvement.

As some of the interventions promoted pupil choice, two focus groups were undertaken with pupils from each of the three schools post-intervention, following procedures similar to that of the interviews. These focus groups took place in an empty classroom and lasted, on average, 30 minutes (range 23–40 minutes), with three to six pupils participating at any one time [53]. The focus groups all followed a semi-structured topic guide, which discussed, i) what pupils and their classmates thought about CLASP, ii) whether pupils thought anything had changed during participation, iii) if pupils would like CLASP to continue, and iv) whether or not pupils thought the school would continue with their chosen intervention components. Pupils were selected randomly to participate in focus groups following purposive allocation dependent on gender, deprivation (FSM entitlement) and participation in the interventions (identified from attendance collected through direct observations). Those pupils who did not participate in optional interventions, such as alternative activities, were included in the focus groups to understand reasons underpinning lack of engagement. For the daily classroom refresher intervention, pupils were selected at random from all those who had provided consent to participate. Engagers and non-engagers participated together in the focus groups in order to promote more organic discussions regarding facilitators and barriers. Participants were selected via stratified randomisation to ensure equal numbers.

**Quantitative measures.** *Physical activity*. Physical activity was objectively measured at 100 Hz using the GENEA © accelerometer (GENEActiv, Unilever Discover, Sharnbrook, Bedfordshire, UK), a triaxial, ± 6g seismic acceleration sensor, which has been previously validated for use in children [54]. Monitors were placed on the non-dominant wrist, to be worn 24 hours per day, for seven full days, including while sleeping and during water activities. The GENEActiv has excellent criterion validity in both adults (r = 0.86) and children (r = 0.91) when worn on the left wrist, mainly classified as the non-dominant wrist [54, 55].

*Aerobic fitness*. Fitness was measured through the well-validated 20m-shuttle test, using methodology described by Leger et al. [56].

*Intervention dose and fidelity*. Schools maintained records of the number of sessions that took place during the intervention to record dose. Coaches were asked to complete attendance records to assess engagement with sessions. Direct observations of sessions (n = 3) in all three schools were undertaken (by DC) throughout to assess fidelity and attendance at sessions.

## Data analysis

Interviews and focus groups were analysed through schema analysis, fully described elsewhere [57]. Briefly, each researcher (DC & CT) developed schemas, or small sections of text detailing a common thought, from the transcripts independently. These schemas were coded by topic, such as 'coach enthusiasm', before the second researcher verified the schemas coded by the first researcher. No *a priori* hypothesis was determined and commonalities across schemas were collated to form themes, allowing the key thoughts from participants to be identified from the data. Schema analysis is an equalising method, with all researcher views pertinent and considered, that ensures validity of the working approach through group understanding [58]. Although agreement between researchers was high, any discrepancies were discussed until a consensus was reached. Qualitative and quantitative data were integrated using the triangulation protocol for mixed-methods research [59]. The data were initially analysed

separately, as described above, and then combined to look for areas where similarities or discrepancies in the findings occurred. In addition to the quantitative outcome evaluation, a process evaluation was conducted, guided by the RE-AIM framework [60]; a common model used to evaluate implementation [61]. This detailed intervention fidelity, changes in pupil engagement, and qualitative views pertaining to maintenance.

The raw GENEActiv data was downloaded and the .bin files converted to 60-second epoch .csv files using GENEActiv PC software version 2.1. The 60-second epoch data files were entered into an open-source Excel macro (v2; Activinsights Ltd.) in order to eliminate sleep time [62]. Non-wear was assessed through previously described methodology [63]. KineSoft software (version 3.3.75; KineSoft, Loughborough, U.K.) was used to produce a series of standardised accelerometry outcome variables following procedures similar to those described by Esliger and Tremblay [55] and Esliger et al. [64]. To be included in the analyses, participants had to meet the wear-time criteria of 60 minutes on any three days [65]. Validated acceleration magnitude cut-points were used to classify activity intensity (min·day$^{-1}$) [54].

Paired t-tests were conducted to assess changes in MVPA, sedentary time and fitness from baseline to post-intervention and follow-up. Paired t-tests were used due to unequal numbers of observations between time-points and the low sample size that would have resulted from requiring observations at all three time-points. Additionally, in this instance, the assumption that compound variance would not differ could not be guaranteed. Preliminary analyses to ensure normal distribution of data were completed prior to all further analyses. STATA V.12.1 (STATA, Texas, USA) was used for all statistical analyses and statistical significance was set at p<0.05 throughout.

## Results

The results section will firstly outline the choices of intervention components by school and the reasons for this selection. The outcome and process evaluation results will then be formatted in accordance with the RE-AIM framework; reach, effectiveness, adoption, implementation and maintenance. In this instance, adoption will be presented prior to effectiveness to provide clarity due to the nature of the intervention.

### Intervention component choice and reasons for selection

The intervention components chosen per school were; School A–Alternative activities (Street dance and basketball), School B–Alternative activities (Street dance) and Parent and Child afterschool sessions (Family Boxfit), and School C–Daily Classroom Refreshers (Fig 2).

Although all three schools were provided with autonomy over intervention choice, the three headteachers exercised their autonomy in very different ways, and opted for different approaches to tackle their school's physical activity needs (Fig 2). During the initial interviews, headteachers from two schools (A and B) mentioned that they strived to be democratic in their approach and discussed the options with respective deputies or P.E. co-ordinators. However, Headteacher C took a more autocratic approach.

School A chose alternative activities, as the headteacher believed these were something they could not offer themselves as a school, though expressed a preference for allowing pupils to choose which specific activities were implemented. School B also chose alternative activities, in addition to parent and child activity sessions, as the headteacher wanted to address and improve parental engagement. School B was also keen to honour student and parental choice in the selection of activities. Pupils were administered surveys by researchers prompting selection of varying types of sports or activities, and parents were invited to a coffee morning at the

| | School A | School B | School C |
|---|---|---|---|
| **Intervention chosen** | **Alternative activities** (Street Dance & Basketball) | **Alternative activities** (Street Dance) **Parent and child activity sessions** (Family Boxfit) | **Daily classroom refreshers** |
| **Level of choice** | Headteacher ↓ Children (SD/B) | Headteacher ↓ Children (SD/FB) | Headteacher |
| **Implementation dose** | SD 8/11 sessions B 6/11 sessions | SD 8/11 sessions FB 8/11 sessions | 4-5 x a week |
| **Facilitators** | • Coach enthusiasm and experience (SD) • Inclusivity (SD) • Performance element (SD) | • Different to curriculum • Siblings could attend together (FB) • Performance element (SD) | • Enthusiastic school lead • Whole-class intervention • Improved concentration |
| **Barriers** | • Coach inconsistencies • Competition for afterschool time • Parental attitudes | • Competition for afterschool time • Lack of within school leader • Coach inexperience/ discipline • Parental time | • Competition for in school time • Curriculum pressures • Teacher time |
| **Sustainability** | ✓ Street Dance ✗ Basketball | ✗ Street Dance ✗ Family Boxfit | Daily classroom refreshers ✓ (Limited) |

**Fig 2. CLASP implementation schematic.** Legend: The down arrow shows where the headteacher, teacher and children had a choice in the intervention, whereas for the school C, the headteacher made the choice (SD = Street Dance, B = Basketball, FB = Family Boxfit).

school to discuss different activity types. Leaflets notifying the days and times of the sessions taking place were sent out to parents and pupils.

The headteacher from school C decided on a curriculum-based approach. In this instance, the pupils had no choice over the intervention component. Indeed, school C chose daily classroom refreshers as the headteacher believed this approach was advantageous for concentration, behaviour and academic achievement and would '*capture all children as opposed to a haphazard few that would attend an out of school activity*'. School ground constraints, previous

unsuccessful experiences, litigation risk and high numbers of existing afterschool activities meant other options were less attractive across all three schools.

## Reach

The reach of the interventions differed greatly between schools. School C, which had daily classroom refreshers, engaged 100% of pupils as this was undertaken during usual classroom sessions. For schools A and B, attendance fluctuated greatly between voluntary afterschool sessions. Attendance records were completed sporadically, leading to insufficient data capture, and therefore this data could not be quantified with any certainty.

## Adoption

Nine schools were contacted initially with three expressing an interest. These three schools (FSM 9%-53%, mean 34%) demonstrated a 33% adoption; slightly lower than the 47% adoption of a recent similar physical activity intervention study [66]. Reasons for non-participation from the other six schools included a new headteacher who was not involved in the first phase of CLASP [14, 15], and a headteacher who was currently undergoing health issues. No information was provided as to why the other four schools did not respond.

## Effectiveness (physical activity, sedentary time and fitness)

Of the 85 individuals who participated in the study, 72 pupils across the three schools met the accelerometer wear-time criteria and were included in the analyses. Due to the paired t-test analysis, if results were present for only one time point the data was removed from the analysis.

When MVPA was stratified by school, all three schools showed a positive trend between baseline and post-intervention (Table 2), though this was only significant for school C. There were significant increases in MVPA from baseline and follow-up for all three schools. Similarly, sedentary time reduced in all three schools at post-intervention, with schools A and C demonstrating a significant decrease. At follow-up, significant decreases in sedentary time of 118, 118 and 100 min.day$^{-1}$ were observed for schools A, B and C, respectively.

Fitness improved significantly for schools A and C between baseline and post-intervention, whereas only small increases in fitness were reported in school B. Interestingly, only school A continued to demonstrate an increase at follow-up. Fitness measures in schools B and C at follow-up were comparable to baseline.

## Implementation

The implementation type, levels of autonomy and the dose of sessions delivered for all three schools is presented in Fig 2, in addition to implementation facilitators and barriers expressed by headteachers, teachers and pupils.

**Dose and fidelity.** *School A (alternative activities–street dance and basketball).* The street dance group completed 8 out of 11 sessions, including an assembly performance, and 6 of 11 basketball sessions were delivered. Basketball sessions were mainly cancelled as a result of inconsistent attendance by the coach (four sessions) and a clash with school parents' evening. Cancellation of street dance was also due to a clash with parents' evening and school transition periods to high school. The headteacher noted attendance started high for street dance, but decreased with time, whereas participation in basketball was lower at the outset but increased steadily throughout, due to word of mouth.

**Table 2. Changes in MVPA, sedentary time and fitness per school between baseline, post-intervention and follow-up.**

| | | School A | School B | School C |
|---|---|---|---|---|
| **MVPA** | | n = 20 | n = 11 | **n = 23** |
| | Baseline | 99.0 (31.4) | 105.2 (48.0) | **99.9 (30.7)** |
| | Post-intervention | 107.2 (39.4) | 114.2 (43.4) | **117.0 (36.3)** |
| | Difference | 8.3 (24.6) | 9.0 (50.5) | **17.0 (25.9)** |
| | (95%CI) | -19.8 to 3.3 | -42 to 25.0 | **5.8 to 28.2** |
| | | **n = 18** | **n = 11** | **n = 19** |
| | Baseline | **97.3 (32.6)** | **103.6 (42.2)** | 97.1 (31.2) |
| | Follow-up | **144.8 (60.8)** | **147.9 (33.6)** | 135.3 (49.4) |
| | Difference | **47.5 (54.5)** | **44.3 (41.8)** | 38.3 (30.5) |
| | (95%CI) | **20.4 to 74.6** | **16.2 to 72.4** | 23.6 to 53.0 |
| **Sedentary Time** | | **n = 20** | n = 11 | **n = 23** |
| | Baseline | **687.5 (96.9)** | 706.1 (123.0) | **707.7 (50.4)** |
| | Post-intervention | **616.7 (72.7)** | 677.2 (71.1) | **643.1 (103.0)** |
| | Difference | **70.8 (78.8)** | 28.9 (83.9) | **64.7 (106.2)** |
| | (95%CI) | **33.9 to 107.7** | -27.5 to 85.2 | **18.7 to 110.6** |
| | | **n = 18** | **n = 11** | **n = 19** |
| | Baseline | **692.5 (100.5)** | **701.5 (118.1)** | **706.3 (53.0)** |
| | Follow-up | **573.6 (148.4)** | **582.8 (75.4)** | **606.2 (99.4)** |
| | Difference | **118.9 (145.5)** | **118.7 (99.1)** | **100.1 (83.7)** |
| | (95%CI) | **46.5 to 191.3** | **52.2 to 185.3** | **59.7 to 140.4** |
| **Fitness** | | **n = 20** | n = 16 | **n = 24** |
| | Baseline | **31.1 (13.5)** | 25.9 (13.7) | **38.6 (14.6)** |
| | Post-intervention | **39.8 (17.6)** | 27.3 (12.5) | **43.2 (15.8)** |
| | Difference | **8.7 (14.6)** | 1.4 (12.7) | **4.6 (8.4)** |
| | (95%CI) | **1.9 to 15.5** | -8.2 to 5.3 | **1.0 to 8.1** |
| | | **n = 18** | n = 15 | n = 25 |
| | Baseline | **28.8 (14.0)** | 25.9 (13.7) | 39.2 (13.3) |
| | Follow-up | **39.1 (18.7)** | 29.2 (10.0) | 38.7 (14.7) |
| | Difference | **10.3 (15.9)** | 3.3 (7.6) | -0.5 (8.9) |
| | (95%CI) | **2.4 to 18.2** | -7.5 to 0.9 | -3.2 to 4.2 |

Data represented as Mean (SD), unless otherwise stated. Post-intervention refers to three months post-baseline (April) and follow-up refers to six months post-baseline (July). Bold = achieves significance (p<0.05).

*School B (alternative activity & parent and child activity)*. Street dance completed 8 out of 11 sessions but did not manage to undertake the performance. Reasons for cancellations included a clash with parent's evenings, school strikes and availability of coach. Again, the headteacher reported attendance started high for street dance but decreased steadily throughout. Parent and child afterschool boxfit sessions started 2 weeks after the other sessions due to initial lack of interest (8 out of 11 delivered). A few parents participated in the first sessions, but direct observations of sessions found these quickly became pupil-only sessions. However, these sessions still promoted family engagement as siblings attended together and parents verbally interacted during sessions.

*School C (daily classroom refresher)*. Daily activity energisers were reported by the teacher as being completed an average of 4/5 times a week (less on busier weeks). When used at times that were least disruptive, it was felt they aided pupils' concentration and helped break up monotonous periods during the school day.

**Factors affecting intervention implementation.** Headteachers and teachers reported a number of factors which influenced the delivery of the chosen interventions, and pupils reported factors which influenced their engagement or disengagement. These qualitative insights provide further understanding of the difficulties these schools faced when implementing new interventions, including; coach consistency, enthusiasm and session delivery, alignment with existing curriculum, competition for time, the need for a school lead to champion the project, inclusivity, parental attitudes and autonomy.

*Coach consistency, enthusiasm and session delivery.* The impact of the specific coach, and their approach to the sessions, was highlighted as influential, with enthusiasm, confidence and consistency all key factors in both engaging the pupils and maintaining delivery of the sessions. Basketball sessions were less structured, as the coach was unable to attend every session. The headteacher (school A) believed these inconsistencies caused the children to lose interest and believed, *'the take up wasn't as good with the basketball but I think that was more to do with sometimes the coach was letting them down and I think, you know what children are like. . .if things are not completely consistent they just give up don't they?'.*

The headteacher of school A felt that *'the street dance was more successful than the basketball, but that was more to do with I think the enthusiasm of the coach really, so. . . we're going to continue to use them as a coach into September'*. This headteacher perceived the enthusiasm from both the street dance coach, coupled with support from the Head of Physical Education (P.E.), to be a key driver to effective implementation. The pupils from school C also mentioned the enthusiasm of the teacher as a factor, stating that daily classroom refreshers at the start were much better. Pupils stated that initially the daily classroom activities varied considerably, but after a while, the same activities, mainly running, were repeatedly used, causing some repetition and reluctance to participate. '*At the beginning we were doing it with balls and everything and then like every day we'd just do running'*. This was predominantly reasoned by pupils to result from a lack of teacher time to plan activities.

Bad behaviour was detailed by the external coach as having a distinct influence in school B, which became more of an issue as sessions progressed, especially with the girls. This had a knock-on effect on attendance as the focus was taken away from the activity itself, making it less enjoyable for all. One pupil stated, '*I think everyone quit, I think everyone quit because it was just like a lot of arguments wasn't there?'*. Moreover, the coach reportedly found it difficult to differentiate for all abilities and engagement levels, and reported it was hard to teach sometimes because some pupils attended predominantly because '*their friends had come along'*, which led to '*some being engaged, some not'*. The accumulation of these issues meant unfortunately school B could not proceed with the street dance performance as the pupils were not prepared enough. However, the headteacher from school A believed the performance helped '*create an event'* and amplified enthusiasm.

*Alignment with existing curriculum.* Initial motivating factors for headteachers selecting intervention components (schools B and C) included the perception that the project provided a great opportunity for pupils to participate in new activities whilst contributing towards health and well-being elements accountable to the schools' inspectorate body. Conversely, one teacher from school B thought street dance and boxfit had managed to engage those disinterested with P.E., mainly because it was so different from the current prescriptive P.E. curriculum. This teacher commented, *'they know what sort of thing they're gonna [sic] be doing as they go through school in PE, but it was so different, so it got their attention'.*

Further positives include the fact that daily classroom refreshers did not require any special equipment and were not particularly time-consuming, thus not taking time away from core curriculum components. However, the teacher delivering the classroom refreshers (school C) found the project difficult to consistently implement on a day-to-day basis due to curriculum

time pressures. This teacher stated the activity sessions were, *'just another project to fit into the day'*.

*Competition for in-school and afterschool sessions.* The headteacher from school A believed that, '*if we were running this [street dance] as part of an enrichment activity when they were all in school, they'd be fighting to get onto it'*. Whereas, afterschool sessions rely on children to be motivated enough to stay behind after school. Some children from this school (A) expressed a desire to join as many clubs as possible to alleviate the usual boredom experienced after school in the house. However, others who didn't engage alluded to the competitiveness for time post-school due to clashes with other activities or wanting to spend time with their friends, thus they were influenced by who else attended afterschool sessions.

This competitiveness for afterschool time was reinforced in school B, as some boys who did not attend mentioned family boxfit clashed with their running club. One pupil even asked, *'can we get, change the box fit on Wednesday? Because loads of people need to go to. . .athletics'*. Nonetheless, the headteacher explained that afterschool sessions ran every day so would have clashed regardless of day of the week.

*School lead.* Assigning a designated teacher to promote activities, and chase up children who did not attend, was suggested by both the deputy headteacher and class teacher from school B as one improvement to further enhance attendance. The class teacher remarked that, *'pupils often attend sessions more to appease the teacher than actually wanting to do the activity'*, so this approach may help raise attendance initially, but it is unclear what effect this would have on maintenance. The deputy headteacher remarked that it was imperative the *'right kind'* of teacher was assigned to street dance or boxfit sessions, otherwise this would negate the intended effect. This was evidenced further in school C, as the class teacher had a high level of expertise regarding physical activity, which the pupils saw as a positive. Furthermore, the intervention in school A was led directly from the headteacher, who fully embraced a whole school engagement approach to implementation by including key members of staff in the initial discussions, with the enthusiasm for the project then disseminating throughout the whole school.

*Inclusivity.* When interviewed at follow-up, the Head of P.E. and headteacher from school A favoured street dance's non-competitive nature and the focus on teamwork, meaning it was more inclusive and attracted those normally disengaged with physical activity. This was further endorsed by the Year 5 teacher from school B, who commented that, *'there were some children who took part that I didn't think would. . .on the yard they don't join in with football, basketball, anything like that, they just sort of keep to themselves, so for them to be included in a group exercise was a big deal'*. Conversely, one headteacher reported the competitiveness of basketball was viewed as off-putting by pupils in school A. Pupils from school C discussed in focus groups that daily classroom refreshers engaged the whole class, though did note that during periods of extended writing, the sessions could be disruptive. However, the teacher stated that the daily energisers would be best used, *'more for concentration I think . . .especially in primary school they have break time in a morning, they have a break time in the afternoon, and they're always up on their feet moving about the class, so I don't feel that it makes a lot of difference to their healthy lifestyle'*.

*Parental attitudes/time.* Parents' attitudes were perceived as a barrier to afterschool attendance for parent and child afterschool sessions in school B, and this headteacher said that, *'getting our parents to engage sometimes can be quite difficult'*. Parents' own experiences were perceived to have an impact as some, *'didn't have a particularly good experience of school, so even to just get some of the parents in [to school] is a huge thing'*. Pupils from school B listed logistical issues why parents were unable to attend, such as, *'mum and dad are at work'*, or, *'mum works nights so she has to sleep in the days'*. Though others referred to more generic attitudinal factors towards physical activity such as, *'my dad doesn't like to exercise'*, or, *'my mother*

*would think it's a bit ridiculous to pay to get fit whereas we can just like do it on the streets our-selves'*. Pupils preferred the idea of taking part during school time to remove these attitudinal barriers of parents influencing what they chose to do.

*Autonomy.* Initially, all three headteachers were positive about this novel approach, stating, *'It was nice that there was a partnership and exciting that there was something that could be talked about and agreed upon'*. One headteacher (school C) said in an ideal world, schools would be presented with a choice of options then schools would find it easier to adopt a pro-gramme suitable for their needs, as *'everyone can maybe choose something then'*. Conversely, when interventions offer only one project, some schools would say, *'No that's not going to work for us, no thank you'*; limiting rates of intervention adoption. With all schools having differing agendas, the headteacher from school A believed they were best placed to understand the indi-vidual needs of their school and how to most effectively address these by choosing an interven-tion that best suits them.

Whilst school staff enjoyed the opportunity to select their own interventions, in some instances there was discordance between pupils' and headteachers' tastes. Interestingly, all schools noted that if they were to participate in the CLASP intervention again, they would pro-vide pupils with greater autonomy and allow them greater ownership, rather than just the school leadership team. The deputy headteacher from school A reported that permitting pupils choice over the types of activities implemented within CLASP was definitely valuable and helped those pupils usually disengaged with P.E. to engage with physical activity. The school was able to align this approach with its existing policies for promoting pupil voice. This increased the ownership for pupils, which generated an element of accountability for missed sessions and helped maintain attendance levels. Additionally, due to existing practices in schools, pupils opined that it would be unfair if they had no choice in the matter. The general consensus from these pupils was that, *'children like choosing'*, and that asking children what they wanted to do was the best option to increase physical activity, as opposed to headteachers pre-selecting sports or activities at random for pupils to try.

## Maintenance

Assessing the maintenance of these projects was a key focus of this study. As reported earlier in the effectiveness section, favourable changes in MVPA, sedentary time and fitness were observed, most of which were sustained at the three-month follow-up (Table 2). Only one of the three schools, school A, maintained sessions after the mandatory intervention period of three months. The key difference was a whole school enthusiasm for the intervention, from the headteacher and head of P.E., all the way down to the pupils. Direct observations found this headteacher was present at the majority of afterschool sessions, demonstrating full engage-ment and enthusiasm for the project. Additionally, the enthusiasm of the street dance coach and the enjoyment of the performance element played a role in sustaining these sessions. Observations in schools B and C found the headteachers rarely attended sessions. The headtea-cher in school C went as far as to say, *'I haven't seen an awful lot of it. . .I've pretty much left it to (the teacher)'*. Although there was class integration in school C, there was only limited main-tenance of the daily classroom refreshers intervention at follow-up. The teacher suggested this was mainly due to the class management benefits, as opposed to health benefits, and stated, *'if I see that they're finding a task difficult where they really have to focus, or they're finding it hard to concentrate, that's when I'd take them out'*. Therefore, the daily classroom refreshers were implemented on an ad hoc basis and much less often than the once-a-day employed during the intervention.

## Discussion

This study aimed to involve headteachers in the developmental stages of school-based health interventions to allow them greater autonomy and explore how this influenced adoption, effectiveness, implementation and maintenance. The CLASP intervention demonstrated that providing headteachers with a choice of physical activity projects was a positive approach to the adoption of a school-based intervention as this was viewed as a more collaborative approach to working. Mixed results were reported for the effectiveness, implementation and maintenance of an autonomous model. However, contributing influential factors were similar to those reported in more traditional school-based health interventions, such as a lack of time and existing curriculum pressures.

Headteachers appreciated the opportunity for greater autonomy regarding interventions during the developmental, adoption and implementation stages; concurring with previous research suggesting that engaging key stakeholders during initial stages improves intervention implementation [67]. The increased autonomy given to headteachers during this study allowed them to select intervention components that best aligned with their current priorities and personal values; an important guideline for developing complex interventions [17] and a key recommendation of the 'Health Promoting Schools' agenda [18]. The choice of five research-informed physical activity interventions provided greater adaptability, enabling each headteacher to select a project that best suited their school's needs, as opposed to traditional, standardised intervention styles. The selection of different intervention components amongst the three schools demonstrates choice is both desired and warranted.

Curriculum pressure and the need to prioritise core subjects, such as literacy and numeracy, were influential factors in headteachers' decisions regarding intervention choice. Afterschool sessions proved popular from a headteacher perspective, as they were less burdensome on schools in terms of implementation. Time and curriculum pressures have regularly been identified as a barrier to the implementation of traditional school-based physical activity initiatives [15, 68] and this does not appear to be specific to an autonomous approach. Interestingly, the only school not to select an afterschool session implemented daily classroom refreshers that were designed to engage the whole class, even though this was during curriculum time. This was due to the headteacher's understanding of the positive impacts of physical activity breaks on concentration, learning and behaviour; concurrent with beliefs widely reported in the literature [22, 23, 69]. This was believed to be a more inclusive approach that would engage all pupils, as opposed to only those motivated and able to stay behind for afterschool sessions. This is consistent with previous research detailing that afterschool sessions would need to be very attractive in order to have high engagement [36]. Attendance at afterschool sessions is known to be influenced by enjoyment [70] and the provision of transportation home after the session [71]. Therefore, the headteacher believed they would be improving health and academic achievement for the class as a whole, rather than only the few; a universal approach consistent with the population strategy detailed by Geoffrey Rose [72]. This difference in approaches suggests that when offered a choice, headteachers may prioritise interventions which fit best around existing pressures, such as curriculum pressure, as opposed to those which demonstrate greater effectiveness but are more burdensome for schools to implement. Therefore, it is advisable to offer whole school, evidence-based choices which have shown potential for effectiveness, ideally in partnership with capacity-building for schools to aid delivery.

It is noteworthy that only one school (A) opted to fully maintain their intervention, and the headteacher identified whole school support as an important facilitator in this maintenance. This is in accordance with the 'Health Promoting Schools' agenda which promotes whole

school or class integration of an intervention [18]. Class integration was also evident in the partial maintenance of the inclusive daily classroom refreshers in school C.

One headteacher believed a second key facilitator for maintenance was the street dance performance, as this promoted street dance to non-participating pupils and helped increase excitement about the sessions; a known facilitator for school-based interventions identified by headteachers [15]. Headteachers also reported that the member of staff chosen to promote the activity needed to be an appropriate teacher who could motivate pupils to participate. Social support from teachers has previously been identified as a significant mediator in improving physical activity levels in children [73]. The importance of this teacher-pupil interaction highlighted throughout CLASP warrants further research to fully understand the effects on both motivation and physical activity engagement.

The key concept explored through this study was increased autonomy of schools, and this proved influential during the implementation stage of the intervention through promoting more of a partnership approach between the headteacher and researchers. Interestingly, only one headteacher fully discussed the intervention option with their teachers before implementation. Whilst it is important to engage the headteachers initially, buy-in is also required across the whole school. Previous research has discussed the discordance between administrators' and teachers' views on health-based interventions and the impacts on implementation [74]. Moreover, a supportive school climate has been identified as a key factor for effective implementation [68].

Another key recommendation from headteachers and pupils, for improved implementation of school-based interventions, was the inclusion of pupils in the consultation process. Schools that utilised this approach noted multiple benefits, such as improved engagement and the promotion of pupil voices, and planned to introduce this aspect at an earlier stage for future projects. At follow-up, the only school who did not incorporate the views of pupils, expressed an interest in exploring this in the future. Previous research demonstrated children are rarely included in the design and implementation of projects [75]. The recommendations from headteachers in this study demonstrated the approach works well within a primary school setting, in the context of physical activity, and would benefit from further exploration in future school-based interventions.

## Strengths and limitations

The mixed-methods approach used throughout the CLASP project was a key strength. The quantitative outcomes allowed an insight into the effects of the interventions on changes in physical activity, sedentary time and fitness. Furthermore, the extensive qualitative work, with multiple recipients of the intervention, provided a rich, contextual understanding of the acceptability and fidelity of the intervention and the mediators underpinning the quantitative changes seen in pupil health behaviours [49]. Whilst all three intervention choices reported favourable changes to MVPA, sedentary time and fitness, these results should be interpreted with caution. Given the timing of the intervention (January through to July), the influence of seasonal variation (Winter to Summer) cannot be precluded [76]. Additionally, school A mentioned that participating in this study increased awareness of pupils' physical activity levels, and the impact of additional school interventions cannot be separated out. The lack of feedback from all the coaches at follow-up was also a limitation. Moreover, the sporadic recording of attendance of afterschool sessions prevented reach being calculated with any certainty. Future studies would benefit from the use of age- and sex-matched comparison schools to provide additional insights into the results reported here. Finally, the utilisation of the RE-AIM framework helped guide a more rigorous outcome and process evaluation [60], allowing

greater insights from a proof-of-concept perspective. Furthermore, the focus on adoption and maintenance, in addition to implementation, was innovative as these aspects have been identified as under-researched areas within this field [61].

## Conclusions

Headteachers perceived that being provided greater autonomy resulted in much more of a partnership approach to school-based interventions and welcomed the idea for future interventions. However, mixed results were reported for the effectiveness, implementation and maintenance of the interventions. Nonetheless, headteachers highlight that involving pupils in the decision-making process and promoting a positive school environment were key factors for enhancing engagement. Promoting inclusive physical activity projects, with a consideration of existing curriculum pressures, aided implementation for headteachers. Overall, this mixed-methods study provides valuable insights about autonomous approaches that could inform further development, implementation and maintenance for future interventions.

## Acknowledgments

All authors would like to thank the staff at the participating schools for their co-operation during the study, and also the pupils for their views and opinions as well as participation.

Dedication: This work was designed with the late Professor Non Thomas who is greatly missed by us all.

## Author Contributions

**Conceptualization:** Danielle L. Christian, Sinead Brophy.

**Data curation:** Danielle L. Christian, Charlotte Todd, Kelly A. Mackintosh, Sinead Brophy.

**Formal analysis:** Danielle L. Christian, Charlotte Todd, Kelly A. Mackintosh, Frances Rapport, Sinead Brophy.

**Funding acquisition:** Sinead Brophy.

**Methodology:** Danielle L. Christian, Charlotte Todd.

**Project administration:** Danielle L. Christian, Charlotte Todd.

**Supervision:** Jaynie Rance, Gareth Stratton, Sinead Brophy.

**Writing – original draft:** Danielle L. Christian, Charlotte Todd, Sinead Brophy.

**Writing – review & editing:** Danielle L. Christian, Charlotte Todd, Jaynie Rance, Gareth Stratton, Kelly A. Mackintosh, Frances Rapport, Sinead Brophy.

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
