## [Decision Letter · Decision Letter 0]

9 Jan 2020

PONE-D-19-31826

Involving the headteacher in the development of school-based health interventions: A mixed-method outcome and process evaluation using the RE-AIM framework.

PLOS ONE

Dear Dr. Christian,

Thank you for submitting your manuscript to PLOS ONE. After careful consideration, we feel that it has merit but does not fully meet PLOS ONE’s publication criteria as it currently stands. Therefore, we invite you to submit a revised version of the manuscript that addresses the points raised during the review process.

We would appreciate receiving your revised manuscript by Feb 23 2020 11:59PM. To enhance the reproducibility of your results, we recommend that if applicable you deposit your laboratory protocols in protocols.io, where a protocol can be assigned its own identifier (DOI) such that it can be cited independently in the future. For instructions see: http://journals.plos.org/plosone/s/submission-guidelines#loc-laboratory-protocols

We look forward to receiving your revised manuscript.

Kind regards,

Lawrence Palinkas

Academic Editor

PLOS ONE

Reviewers' comments:

Reviewer's Responses to Questions

**Comments to the Author**

1. Is the manuscript technically sound, and do the data support the conclusions?

Reviewer #1: Partly

Reviewer #2: Partly

2. Has the statistical analysis been performed appropriately and rigorously? 

Reviewer #1: Yes

Reviewer #2: Yes

3. Have the authors made all data underlying the findings in their manuscript fully available?

Reviewer #1: Yes

Reviewer #2: No

4. Is the manuscript presented in an intelligible fashion and written in standard English?

Reviewer #1: Yes

Reviewer #2: Yes

5. Review Comments to the Author

Reviewer #1: Lines 72-74 – The Action Schools! BC framework is cited as the example intervention but would be helpful to clarify at first citation or in Table 1 (line 106) if any of the intervention components described in this intervention overlap with or were a featured component from Action Schools! BC. Asked slightly differently, I don't see any of the Action Schools! BC journal citations in the supporting evidence column. Does this mean the components from that intervention are not included in the intervention options for this research project?

Line 85 “Deprivation was classified…” – is this phrase referring to exclusion or inclusion criteria?

Line 106 – there are a number of peer-reviewed journal articles that highlight the impact of classroom-based physical activity. The first reference below would likely inform this article; it is more current than reference 24.:

• Watson A et al. Effect of classroom-based physical activity interventions on academic and physical activity outcomes: a systematic review and meta-analysis. Int J Behav Nutr Phys Act. (2017)

• Carlson JA et al. Contextual factors related to implementation of classroom physical activity breaks. Transl Behav Med. (2017)

• Donnelly JE et al. Classroom-based physical activity, cognition, and academic achievement. Prev Med. (2011)

Line 106 - Question – In Table 1, “In The Zone” project, it was noted there was a “training workshop for lunchtime supervisors.” Was teacher training provided for the other 4 interventions? Or, were there "overviews" for teachers regarding how the other 4 interventions should be implemented with fidelity (even if they were led by a coach) to help with sustainability?

Line 112 – are the UK sites operating in a year-round school setting? (Follow-up in July would be almost impossible in the U.S. school systems so indicating the schools operate year-round would help a reader).

Lines 134-135 – RE: “The two other coaches declined invitation to participate in an interview due to work commitments.” Questions: Were these coaches paid by the study to participate as physical activity (PA) leaders only? Were they employed by the school (e.g. as physical education teachers) or external PA leaders/coaches, for example, from a local fitness or recreation center? Was the interview a condition of engagement as a coach at the front end of the study? Did they receive a stipend for “coaching” and providing feedback on the intervention?

Line 144- 147 – notes pupils were selected randomly and then “Those pupils who did not participate in optional interventions, such as alternative activities, were included in the focus groups to understand reasons underpinning lack of engagement.” Questions: 1) Were those pupils who did not participate in the interventions in the same focus groups as those who did participate? 2) Did those pupils who did not participate have consent to participate and chose not to, that is, was participation optional by the teacher?

Line 233 – Is it possible to include the range of pupils / parents that were reached for the after school activities (even if attendance records were sporadic)? Given intervention strength is often determined by dose x reach this seems to be a disclosure gap.

Line 298 - Coach consistency, enthusiasm and session delivery: It would be helpful for a reader to have one or two opening statements that basically summarize what supported implementation and what deterred implementation and or pupil participation.

Lines 378-379 - it would be helpful to have a sample testimonial statement for this text: "the idea of using

daily energisers as an active break when concentration waned appealed as a future approach"

Line 380 - Parental recruitment strategies are not outlined in the paper. Understanding what techniques were used would be great. Were paper notices sent home? Were intervention details (time commitment) sent via email or in a voice mail? Were there reminder notices to parents?

Line 392 - Clarity is needed early in the manuscript re: teacher and pupil input on selection of intervention activities. To what extent did the headteachers consult with / get the opinion of other teachers and pupils when they selected the interventions or did they make the decision on their own? Was the same selection process employed at School A, B and C? Line 403 says "Whilst school staff enjoyed the opportunity to select their own interventions, in some instances there was discordance between pupils’ and headteachers’ tastes." This sentence makes it sound like teachers could pick their own strategy but earlier descriptive text seemed to suggest the headteachers selected the interventions.

Line 439 - Encouraging "impact on ...maintenance" seems to be an overstatement given the results reported in the previous section.

Line 502 - Strengths and Limitations: The lack of feedback from coaches at follow-up would seem to be a limitation. The ability of a school to sustain an intervention that requires an external coach would seem to be challenging for schools unless the coach is employed by the school system and paid for his/her time.

Figure 2 - Level of choice - does the down arrow indicate that all levels between the headteacher and the children had a choice of the intervention? There should be an acronym abbreviation in column School A Intervention chosen to define (Street Dance & Basketball [SD/B])

Reviewer #2: Review

With choice-based/autonomous participatory models recommended by the Health Promoting Schools literature this research presents an interesting and relevant implementation evaluation of an approach where schools choose relevant evidence-based PA interventions to implement. The analysis uses a well-accepted public health evaluation framework to explore implementation and outcomes and the research is very comprehensive. The manuscript is well written although the complexity of the evaluation and the amount of data makes the discussion challenging. Following are some revisions that I believe will enhance the manuscript. The discussion is a key area for revision.

Introduction

The discussion includes some reference to the health promoting schools framework and approach however the introduction doesn’t. On page four in the introduction where there is discussion about the “increasing autonomy design of AS! BC” and where a lack of research on autonomous designs is highlighted, the section could be strengthened with the evidence about health promoting schools approaches which, at their most pure, are meant to enhance school level autonomy. The evidence on HPS approaches tends to support focused ‘choice-based’ approaches where schools are constrained to issues like physical activity, healthy eating or mental health. I am not suggesting how to approach this but that the section should be enhanced related to the broader literature on autonomous HPS designed interventions in schools. (Lines 444-445 from discussion could be moved into this section).

Methods

Page 7 Lines 148-151. The approach to talking to engaged and non-engaged students sounds purposive but you state that pupils were randomly selected. How did you ensure that both types of students were present and represented if you used random selection? (perhaps a detail about how balanced randomized selection ended up being in terms of engaging and non-engaging students). I don’t understand randomly selecting in this situation but at least showing the actual situation that occurred through the process to result in both voices being heard and perhaps highlighting in the results how the comments were similar or different across the two types of students?

Results

Page 11 Line 252 – do you mean across all schools (overall when all three schools were analysed together? As on the previous line you said only School C was significant? These two statements seem to contradict.

Discussion

Overall I like the comprehensive approach to the study but the discussion is the component of the manuscript that is in need of quite a bit of work. It needs to be tightened and focused on the actual aims and results of the study. As it is currently written it is suggesting findings and conclusions that aren’t supported by the data.

1. I think the discussion needs to start out with an introduction that re-highlights the aim, lays out an order to the discussion based on the aim of the study highlighted on Page 4 and also is ordered based on the presentation of the results which were ordered based on RE-AIM. This will help with clarity and focus on the results.

2. Ultimately the introduction should also highlight the big messages (conclusions) that you will provide details about in the discussion e.g. this mixed methods study showed mixed results related to the adoption and implementation of autonomous models and highlighted that autonomous models may be influenced by school factors highlighted previously in the literature. (and reiterate in the conclusions) A key introductory message is about mixed results but that the study provides valuable insights about autonomous approaches that could inform further development and implementation of such approaches. You suggest something like this in the limitations when you highlight that the RE_AIM framework allowed for insights from a ‘proof-of-concept’ perspective.

3. The discussion and conclusions need to be more measured and based on the data. I suggest this because the first sentence 437-438 says it was a positive approach but the adoption rates were similar to, or worse than, choice-constrained studies in the literature (so choice didn’t appear to influence adoption), you didn’t really show effectiveness but potential effectiveness (maybe… those results were very difficult to interpret given the lack of information about participating in the interventions so what caused the effect??). However, during the implementation evaluation the principals told you they liked the choice but 2/3 didn’t sustain. These are very mixed results and need to be discussed far more cautiously.

4. You talk about local adaptation in the discussion (Lines 445-449) but you have no evidence that adaptation went on – you didn’t set out to study it, you didn’t collect data about adaptation and you haven’t looked into the adaptation literature. I suggest removing any comments about adaptation and how the approach demonstrates adaptation is both desired and warranted. You provided autonomy/choice. That is the focus of the article.

5. Overall I would suggest that the discussion needs to be cut back a bit ensuring that you aren’t re-presenting results but discussing the results in the context of the literature. For instance you go school by school discussing individual reasons for adoption where this could be a global statement about the different rationales for adoption and how they are similar or different – then compare this to adoption factors outlined in the literature. I believe Story et al have presented data on administrators perspectives to implementing HPS that might prove useful to compare to.

6. I think both the introduction and conclusion should talk about the key messages – e.g. use of mixed methods and the strength in elucidating the complexity of adoption and implementation. I believe that what you have really started to show is that adoption and implementation autonomous or non-autonomous models may be constrained by similar factors that influence adoption and implementation and are common to the broader school context. This is a very important finding, not yet discussed in the literature.

7. Effectiveness results need to be discussed in light of reach. If you have a completely voluntary intervention in two schools but see increases in physical activity across a set of children that volunteer with no control group are you seeing an intervention effect? If it is cross-site comparison it is no surprise that the universal strategy showed a significant outcome in terms of MVPA but the sedentary results are very hard to explain. How are these children less sedentary when no intervention focused on this and only one showed differences in MVPA (did your LPA show changes?)? Is this seasonality? These data have significant limitations that are not even discussed in the limitations section. They need to be.

Limitations

There is a body of implementation literature that shows Principal support is important and choice is important but a head teacher is one individual that works with a number of other professionals in a school to implement. In some jurisdictions the equivalent of the Head teacher doesn’t adopt anything without asking the teachers if they are willing. The limitations of the administrators choosing need to be discussed. There was a theme where one school did take a ‘consultative approach with their staff’.

Page 20 and 21 Lines 460-461 – this head teacher’s opinion is supported by Geoffrey Rose and the population health concept that universal strategies that create small changes in behavior are more beneficial at the population health outcome level than interventions targeted at those at highest risk. You could add a comment to this effect. However, above I have recommended talking more succinctly and globally about the different rationale for adoption and how interventions may need to offer multiple reasons to Principals to encourage adoption depending on their context and beliefs and that other factors such as staff transitions, illness etc. influenced adoption.

Line 452 –systematic reviews on PA implementation in schools show that curriculum pressure and competing demands are key barriers to school PA implementation. When you are discussing the factors that guided what intervention was adopted it should be placed in the context of that literature. The fact that two of three strategies selected were based on avoiding these barriers and required an external coach is critical. Thus… it may be that when there are no policy drivers a majority of school administrators will select an option that is potentially not as effective nor guaranteed to reach the whole school body, negating the power of the universal school-based approach. In fact your evidence further convinces me of the need for policy interventions accompanied by constrained autonomy interventions and capacity-building to ensure school PA delivery. I don’t expect you to go out on a limb and say this but I am just providing an alternative interpretation of your data to perhaps highlight the need for a more measured interpretation.

Page 23 Lines 518 – involving pupils in the decision-making process, etc. were not (based on your data) key factors in improving adoption and maintenance. They were key factors suggested by stakeholders as potentially contributing to adoption and maintenance. As well, do you actually mean contributing to implementation and maintenance in this sentence? Adoption is the school adopting the autonomous approach (participating) or the head teacher adopting a certain strategy. It is important to be consistent with your language across the document.

Conclusions

I feel that the conclusions currently misrepresent your findings. These need re-writing once the discussion is re-focused and the new approach to the discussing the results is adopted. These should mirror key statements made in the introduction to the discussion and key rationale from the introduction/aims but perhaps offer a recommendation as well (more research needed comparing two approaches etc.)

Perhaps the second sentence on line 517 “However, whilst this aspect engaged headteachers (did it? 30% of those asked participated – How about “whilst headteachers in this study supported the approach”) it was not solely sufficient to support maintenance of the PA strategy chosen nor did it guarantee effective implementation. This may be because teachers who largely are responsible for implementation were not engaged? or… because larger drivers influence adoption and implementation. It appears that autonomous and non-autonomous models may be influenced by similar implementation factors.

I leave it to you to decide on what the key messages are but I do believe that the reframing I suggest makes this a very nice addition to the literature.

6. PLOS authors have the option to publish the peer review history of their article (what does this mean?). If published, this will include your full peer review and any attached files.

Reviewer #1: No

Reviewer #2: No

---

## [Author Response · Author response to Decision Letter 0]

21 Feb 2020

A response to reviewers has been uploaded as a separate Word document.

---

## [Editor Report · Decision Letter 1]

9 Mar 2020

Involving the headteacher in the development of school-based health interventions: A mixed-method outcome and process evaluation using the RE-AIM framework.

PONE-D-19-31826R1

Dear Dr. Christian,

We are pleased to inform you that your manuscript has been judged scientifically suitable for publication and will be formally accepted for publication once it complies with all outstanding technical requirements.

With kind regards,

Lawrence Palinkas

Academic Editor

PLOS ONE
---

## [Editor Report · Acceptance letter]

20 Mar 2020

PONE-D-19-31826R1 

Involving the headteacher in the development of school-based health interventions: A mixed-methods outcome and process evaluation using the RE-AIM framework. 

Dear Dr. Christian:

I am pleased to inform you that your manuscript has been deemed suitable for publication in PLOS ONE. Congratulations! Your manuscript is now with our production department. 

With kind regards,

on behalf of

Dr. Lawrence Palinkas 

Academic Editor

PLOS ONE